# Small RNA Sequencing Analysis of STZ-Injured Pancreas Reveals Novel MicroRNA and Transfer RNA-Derived RNA with Biomarker Potential for Diabetes Mellitus

**DOI:** 10.3390/ijms241210323

**Published:** 2023-06-19

**Authors:** Fangfang Mo, Bohan Lv, Dandan Zhao, Ziye Xi, Yining Qian, Dongyu Ge, Nan Yang, Dongwei Zhang, Guangjian Jiang, Sihua Gao

**Affiliations:** 1Traditional Chinese Medicine School, Beijing University of Chinese Medicine, Beijing 100029, China; xiaofang.tcm@163.com (F.M.); lvbohan222@hotmail.com (B.L.); bucmzhaodandan@163.com (D.Z.); 13301049980@163.com (Z.X.); karinaadmire@163.com (Y.Q.); dongyuge@163.com (D.G.); zhdw1006@163.com (D.Z.); 2Kennedy Institute of Rheumatology, University of Oxford, Oxford OX3 7FY, UK; nan.yang@kennedy.ox.ac.uk

**Keywords:** diabetes mellitus, pancreas tissue, microRNA, transfer RNA-derived small RNA, potential target

## Abstract

MicroRNAs (miRNAs) and transfer RNA-derived small RNAs (tsRNAs) play critical roles in the regulation of different biological processes, but their underlying mechanisms in diabetes mellitus (DM) are still largely unknown. This study aimed to gain a better understanding of the functions of miRNAs and tsRNAs in the pathogenesis of DM. A high-fat diet (HFD) and streptozocin (STZ)-induced DM rat model was established. Pancreatic tissues were obtained for subsequent studies. The miRNA and tsRNA expression profiles in the DM and control groups were obtained by RNA sequencing and validated with quantitative reverse transcription-PCR (qRT-PCR). Subsequently, bioinformatics methods were used to predict target genes and the biological functions of differentially expressed miRNAs and tsRNAs. We identified 17 miRNAs and 28 tsRNAs that were significantly differentiated between the DM and control group. Subsequently, target genes were predicted for these altered miRNAs and tsRNAs, including Nalcn, Lpin2 and E2f3. These target genes were significantly enriched in localization as well as intracellular and protein binding. In addition, the results of KEGG analysis showed that the target genes were significantly enriched in the Wnt signaling pathway, insulin pathway, MAPK signaling pathway and Hippo signaling pathway. This study revealed the expression profiles of miRNAs and tsRNAs in the pancreas of a DM rat model using small RNA-Seq and predicted the target genes and associated pathways using bioinformatics analysis. Our findings provide a novel aspect in understanding the mechanisms of DM and identify potential targets for the diagnosis and treatment of DM.

## 1. Introduction

Diabetes mellitus (DM) is a chronic disease characterized by abnormal blood glucose levels and associated with short- and long-term complications [1]. According to the World Health Organization, diabetes is estimated to become the seventh leading cause of death worldwide by 2030 [2]. In order to further study the etiology, pathogenesis and effective treatment, animal models for diabetes are needed in basic experimental research. To date, some animal models have been established to study diabetes. In particular, the rat model induced by HFD and STZ is widely used. Firstly, rats are fed with a high-fat diet to induce obesity, lipid metabolism disorder and insulin resistance [3,4]. Following this, low-dose STZ is injected intravenously or intraperitoneally to damage part of the pancreatic tissue and induce dysfunctional islets and also to increase blood sugar levels [5,6]. Finally, the diabetic rat model is applied [7,8,9]. The advantages of this method are simplicity, less waste produced, and less time necessary for completion. Meanwhile, the diabetic symptoms in the rat model are stable. While this is a suitable approach for basic research in diabetes, the molecular biological changes of these diabetic rats have not been clarified. In particular, the expression profiles, functions and potential roles of small non-coding RNAs in various tissues of diabetic rats have not yet been revealed.

Non-coding RNAs are a class of RNAs that cannot be translated into proteins, but they are involved in the regulation of gene expression and physiopathological processes. MicroRNAs (miRNAs) are small non-coding RNAs that, upon their binding to the 3′-UTR of target genes, inhibit the expression of downstream genes by promoting processes such as transcriptional repression, mRNA degradation or cleavage. Studies have shown that miRNAs are involved in the regulation of a variety of physiological and pathological processes, including glucolipid metabolism, oxidative stress and inflammatory processes, and thus can influence the development of diabetes and its related complications [10,11]. Studies have shown that transfer RNAs (tRNAs), a class of RNAs capable of decoding and translating proteins, are likewise a major source of small non-coding RNAs. tRNA-derived small RNAs (tsRNAs) are divided into two major categories, tRNA-related fragments (tRFs) and tRNA halves (tiRNAs), which are generated from mature or precursor tRNAs and mature tRNA anticodon loops, respectively [12,13,14]. Based on the mapping position of tsRNAs on tRNA transcripts, tsRNAs are further classified into five types: tRF-5, tRF-3, tRF-1, tRF-2 and tiRNA [15,16,17]. In addition, tsRNAs are capable of biological functions, including regulation of cell proliferation, gene expression, RNA processing and DNA damage response [15]. Furthermore, previous studies have revealed that tsRNAs could serve as potential biomarkers, regulatory factors and therapeutic targets of diseases such as osteoporosis and breast cancer [18,19].

Small non-coding RNAs, such as miRNAs and tsRNAs, have attracted extensive attention in terms of different biological functions. However, the expression and the roles of tsRNAs in the pathogenesis of DM remain unclear. Thus, we applied small RNA sequencing (small RNA-Seq) analysis to elucidate the miRNA and tsRNA expression profiles in pancreatic tissue in a DM rat model. In addition, the biological functions of the differentially expressed miRNAs and tsRNAs were predicted by bioinformatics analysis. These results can provide a reference for clinical application and introduce new strategies for the treatment of DM.

## 2. Results

### 2.1. Fasting Blood Glucose Level

As shown in Table 1, the Fasting Blood Glucose (FBG) level in model group was significantly higher than that in normal group, indicating that the DM models were established successfully.

### 2.2. The H&E Staining of the Pancreas in DM Model and Normal Rats

Rat pancreatic tissues were morphologically examined by H&E staining. The microscopic findings indicated that there was damage to the pancreas of DM rats. For example, the number of pancreatic cells was reduced, and the diameter of the islets became shorter. In addition, the islets were structurally disorganized, with vacuoles and swollen nuclei (Figure 1).

### 2.3. Profiling and Characterization of tsRNAs in Normal and Model Groups

To investigate whether the expression of tsRNAs is altered in the pancreatic tissue of diabetic rats, we analyzed the number of subtypes of tsRNA transcripts in the normal and model groups. The pie charts show the distribution of the number of each subtype of tsRNA in the normal and model groups (Figure 2A,B). In addition, Figure 2C,D shows the subtype of tsRNAs derived from different anticodons. In the Venn plot, a total of 324 expressed tsRNAs are shown in the normal and model groups, in addition to 45 and 15 specifically expressed tsRNAs in the normal and model groups, respectively (Figure 2E). Figure 2F shows that the tsRNAs detected in our experiment were all unknown tsRNAs in the tRFdb (relational database of transferred RNA-associated fragments).

### 2.4. Differentially Expressed miRNAs and tsRNAs between Normal and Model Groups

We represented the expression changes of miRNAs and tsRNAs in pancreatic tissues of diabetic rats in the form of heat map, scatter plot and volcano plot (Figure 3A,B). A total of 17 differentially expressed miRNAs were found in the pancreatic tissues of the model group compared with the normal group, including 9 up- and 8 down-regulated miRNAs (Figure 3E). Among them, the most significantly up- and down-regulated miRNAs in the model group were rno-miR-19a-5p and rno-miR-3594-5p, respectively (Table 2). Furthermore, we found that a total of 11 tsRNAs were significantly up-regulated and 17 tsRNAs were significantly down-regulated in the model group compared with the normal group (Figure 3F). Among them, tRF-Arg-ACG-007 (Fold Change ≈ 9.85) and tRF-Ser-GCT-008 (Fold Change ≈ 0.05) were the most significant up/down-regulated tsRNAs (Table 3).

### 2.5. qRT-PCR Validation of Differentially Expressed miRNAs and tsRNAs

We then validated the accurateness and reliability of the sequencing analysis using qRT-PCR. We selected four differentially expressed miRNAs (rno-miR-34a-5p, rno-miR-182, rno-miR-181c-3p and rno-miR-384-5p) and four differentially expressed tsRNAs ( tRF-His-GTG-029, tiRNA-iMet-CAT-001, tRF-Met-CAT-012 and tRF-Lys-TTT-024). As shown in Figure 4, expression profiles of these selected miRNAs and tsRNAs are generally consistent with the results from our RNA-Seq analysis.

### 2.6. Identification of miRNA and tsRNA Target Genes

Since miRNAs and tsRNAs can function by acting on their source genes, we constructed a miRNA/tsRNA-target gene network. First, based on the prediction results, we found that 17 miRNAs and 12 tsRNAs were associated with 5186 and 2521 target genes, respectively (Appendix A). We further retained the target genes with a connectivity degree larger than three to construct the interaction network. As shown in Figure 5, a total of 35 and 19 target genes interacted with up- and down-regulated miRNAs, respectively, while a total of 5 and 17 target genes interacted with up- and down-regulated tsRNAs, respectively. These target genes, including Lpin2, Erf3 and Fkbp5, may be closely related to the role of miRNAs and tsRNAs in the development of DM.

### 2.7. Enrichment Analysis for the Target Genes of Differentially Expressed miRNAs and tsRNAs

To gain additional insight into the potential mechanisms of differentially expressed miRNAs and tsRNAs, we conducted GO and KEGG pathway enrichment analyses. The results of GO analysis of differentially expressed miRNA target genes are shown in Figure 6A,B, in which the most significantly enriched Biological Process (BP) is localization (GO: 0051179), the Cellular Component (CC) is intracellular (GO: 0005622) and the Molecular Function (MF) is protein binding (GO: 0005515). In addition, the target genes of down-regulated expressed tsRNAs were significantly enriched in BP, CC and MF for signaling regulation (GO:0023051), intracellular (GO:0005622) and binding (GO:0005488), respectively. The target genes of up-regulated expressed tsRNAs were mainly enriched in localization (GO:0051179), intracellular (GO:0005622) and protein binding (GO:0005515). For the KEGG pathway enrichment analysis, we found that the most enriched pathways for the target genes of down- and up-regulated expressed miRNAs are leukocyte trans-endothelial migration (Figure 7A) and choline metabolism in cancer (Figure 7B), respectively. In addition, the most predominant enriched pathways for the target genes of down- and up-regulated expressed tsRNAs are butyrate metabolism (Figure 7C) and proteoglycan in cancer (Figure 7D), respectively.

## 3. Discussion

tsRNAs are a class of Dicer-dependent noncoding RNAs that are found in various cell types [20]. Previous studies have shown that tsRNAs have a regulatory role in physiological and pathological processes [21]. Similar to the mechanism of action of miRNAs, tsRNAs can also regulate RNA stability by binding to mRNAs and thus participate in the regulation of physiopathological processes [22].

In order to investigate the expression profiles of miRNA and tsRNA and their function, we performed RNA-seq on the pancreas using a DM rat model. In this study, 17 miRNAs and 28 tsRNAs were differentially expressed between model and normal groups. Several miRNAs are involved in the pathogenesis of DM, including miR-542-5p, miR-34a-5p, miR-182, miR-217-5p, miR-448-3p and miR-384-5p [23,24,25,26,27,28,29,30]. Among these miRNAs, miR-542-5p and miR-182 could lead to potential hyperglycemia and modulate the insulin secretion by targeting FOXO1 [23,31]. miR-34a-5p may induce some expression change in PPARγ, which is a main regulator of lipid and glucose metabolism [32]. In addition, mitochondria play a crucial role in metabolic homeostasis, and alteration of mitochondrial function is a hallmark of diabetes [33]. The research by Roser Farre Garros et al. showed that up-regulated miR-542-5p potentially contribute to muscle atrophy by promoting mitochondrial dysfunction [34]. miR-217-5p could affect mitochondrial function and regulate energy metabolism by targeting SIRT1 [35]. Furthermore, miR-34a-5p and miR-182 might regulate the processes of cell proliferation and apoptosis and also be involved in the modulation of beta cell survival [31,36,37]. Therefore, the roles played by miRNAs may be related to beta cell survival, lipid metabolism, glucose metabolism and mitochondrial function. In our study, miR-19a-5p and miR-3594-5p were the top up- and down- regulated miRNA in the model group as determined by RNA-Seq. However, the role of these miRNAs in the pathogenesis of DM have not been fully investigated. Further experiments will be required to address these questions.

It is known that miRNAs and tsRNAs can regulate gene expression at different levels [38,39]. To better understand the function of miRNAs and tsRNAs, we predicted the target genes of miRNAs and tsRNAs and constructed the interaction network. In the present study, a total of 5186 and 2521 mRNAs interacted with significantly altered miRNAs and tsRNAs respectively. We found that miRNAs may participate in the development of DM by regulating Nalcn, Lpin2 and Hoxa11, which were associated with insulin release, fat distribution and inflammation [40,41,42]. Among the 1719 target genes of the four up-regulated tsRNAs, five genes, including E2f3, Wdr41, Fkbp5, Ubash3a and Nr3c2, were associated with three tsRNAs. Among these genes, Ubash3a was a candidate risk factor in type 1 diabetes [43]. Wdr41 was associated with the development of T2DM [44]. The expression of E2f3 has a role in high-glucose-induced vascular endothelial injury and β-cell quiescence and proliferation [45,46]. In addition, diabetes is usually accompanied by dysregulation of the hypothalamic–pituitary–adrenal (HPA) axis, which is centrally regulated through glucocorticoid (GR) and mineralocorticoid receptors (MR). In the study of Olaf et al., Nr3c2, encoding the mineralocorticoid receptor, was up-regulated in Zucker diabetic fatty rats and implicated in the dysregulation of the HPA axis [47]. In a study of 20 T2DM subjects and 20 non-diabetic subjects, Cherno et al. suggested that Fkbp51 might be a key factor in glucocorticoid-induced insulin resistance [48]. On the other hand, among the 1351 target genes of the eight down-regulated tsRNAs, Irs2 was associated with three tsRNAs. Irs2 is a major component of the insulin/insulin-like growth factor-1 signaling pathway. The disruption of Irs2 impairs both peripheral insulin signaling and pancreatic β-cell function and leads to life-threatening T2DM [49]. In summary, the differentially expressed miRNAs and tsRNAs may participate in the development of DM through regulating the above key genes.

To further reveal the metabolic pathways associated with the target genes of differentially expressed miRNAs and tsRNAs, we performed KEGG pathway enrichment analysis. The results showed that the target genes of miRNAs are enriched in the Wnt signaling pathway, and the target genes of tsRNAs are enriched in the insulin signaling pathway. Wnt proteins, a protein family composed of secreted glycoproteins, are able to regulate a variety of developmental processes [50]. Previous research found that Wnt signaling regulates pancreatic islet proliferation and β-cell proliferation and is crucial for endocrine development [51,52]. In addition, the MAPK signaling pathway and Hippo signaling pathway were associated with both miRNAs and tsRNAs. The MAPK signaling pathway can mediate expression of vascular endothelial growth factor (VEGF), which is one of the major regulatory molecules in diabetes [53]. The MAPK pathway also plays an important role in the inflammation process [54]. Under diabetic conditions, the MAPK signaling pathway is activated and causes the expression of related inflammatory factors [55]. It is closely related to Ras. Hippo signaling is an evolutionarily conserved pathway that critically regulates development and homeostasis of various tissues. This pathway has been proven to play important roles in pancreas development and the regulation of β-cell survival, proliferation and regeneration [56]. Therefore, based on the results of KEGG analysis, we speculated that the altered miRNAs and tsRNAs may influence the development of DM through functioning in pathways associated with inflammation, glycan metabolism and endocrine development.

There are a few limitations of the present study. Firstly, only one timepoint after modeling was selected for study. Therefore, we could not observe the alteration of miRNAs and tsRNAs in the development of DM. Secondly, we selected the pancreas as the study subject. However, due to the complexity of DM pathogenesis, other tissues should be evaluated in future studies. Last but not least, we did not investigate the definitive molecular mechanisms of the differentially expressed miRNAs and tsRNAs, which will be examined in our future projects.

Taken together, this study revealed the expression profiles of miRNAs and tsRNAs in the pancreas using a DM rat model and small RNA-Seq and predicted the target genes and the pathways involved using bioinformatics analyses. Our findings suggest the underlying mechanisms of DM as well as potential targets to the diagnosis and treatment of DM.

## 4. Materials and Methods

### 4.1. Animal Experiment and Tissue Extraction

All experimental protocols were approved by the Animal Care and Management Committee of Beijing University of Chinese Medicine. Male Wistar rats (8 weeks old, Beijing Life River Laboratory Animal Technology Co., Ltd., Beijing, China) were used in this study. All rats were housed in plastic cages at 22 ± 2 °C with a 12-h light/dark cycle. After one week of acclimatization feeding, 20 rats were randomly divided into two groups: the normal group and model group. The rats of the normal group were fed a standard diet (AIN-96G diet, Sibeifu Bioscience Co., Ltd., Beijing, China), and the rats of the model group were fed a high-fat diet (HFD, D12451, 45% fat Kcal%, Sibeifu Bioscience Co., Ltd., Beijing, China) for six weeks. After 12 h of fasting, HFD-fed rats were injected with STZ 30 mg/kg to induce the DM model. Control rats were then injected with an equal volume of sterile sodium citrate buffer. After one week, HFD-fed rats were considered to have DM if fasting blood glucose (FBG) was ≥11.1 mmol/L. Finally, all rats were anesthetized by intraperitoneal injection of pentobarbital sodium anesthesia at 40 mg/kg. After that, rats were sacrificed by exsanguination of the femoral artery. Blood samples and pancreatic tissues were obtained for further experiments.

### 4.2. Histological Examination of Pancreatic Tissues

Pancreatic tissues were fixed with formaldehyde and embedded in paraffin wax. After sectioning, the slices were stained with hematoxylin and eosin (H&E) according to established procedures. Finally, histopathological changes in the pancreatic tissue were observed with an optical microscope (Olympus, Tokyo, Japan).

### 4.3. RNA Isolation and Small RNA-Seq Anaysis

RNA was isolated from the pancreatic tissues of diabetic and normal rats using the Trizol RNA extraction kit (Invitrogen, Carlsbad, CA, USA) according to the manufacturer’s instructions. Each RNA sample was subsequently checked for integrity and concentration using agarose gel electrophoresis and the Nanodrop^TM^ instrument (Thermo Fisher Scientific, Waltham, MA, USA). Total RNA from each sample was ligated to the 3′ and 5′ small RNA aptamers. The cDNA was then synthesized and amplified using Illumina proprietary RT primers and amplification primers. Afterwards, 134–160 bp PCR amplicons were extracted and purified from PAGE gels. Finally, total RNA was quantified using the Agilent 2100 Bioanalyzer (Agilent Technologies, Santa Clara, CA, USA). Sequencing was performed on the Illumina NextSeq 500 system using the NextSeq 500/550 V2 kit (#FC-404-2005, Illumina, San Diego, CA, USA) according to the manufacturer’s instructions.

### 4.4. Data Collection and Analysis

Raw sequencing data generated in the form of Illumina NextSeq 500 was trimmed with cutadapt for sequencing reads with a 5- and 3-adaptor and discarded reads (length <14 nt or length >40 nt) after Illumina quality control. The reads were then compared using bowtie software and miRDeep, respectively, to allow only 1 mismatch to the precursor tRNA sequence and the precursor tRNA sequence. Based on the statistical analysis of the comparison (mapping rate, read length, fragment sequence deviation), we determined whether the results could be used for subsequent data analysis. Finally, the expression profiles of tsRNAs and miRNAs in the normal and model group were obtained.

### 4.5. Validation of Small RNA-Seq Analysis Using qRT-PCR

The sequencing results of selected tsRNAs and miRNAs were validated by qRT-PCR. RNA samples for tsRNAs were firstly pretreated using the rtStar^TM^ tRF&tiRNA Pretreatment Kit (Arraystar, Rockville, MD, USA). Then, cDNA synthesis was performed with the rtStar^TM^ First Strand cDNA Synthesis Kit (Arraystar). Afterwards, the samples were detected using the following thermal cycling parameters on the ViiA 7 Real-Time PCR System (Applied Biosystems, Foster City, CA, USA): initial activation for 10 min at 95 °C, followed by 40 cycles of denaturation for 10 s at 95 °C, and annealing and extension for 60 s at 60 °C. RNA samples of miRNAs were reverse-transcribed with MMLV reverse transcriptase (Epicentre Technologies, Madison, WI, USA) and amplified in the Gene Amp PCR System 9700 (Applied Biosystems). Finally, the expression levels of tsRNA and miRNA were calculated by the 2^−ΔΔCt^ method, and U6 was used as the standard. A list of primers is given in Table 4.

### 4.6. Differential tsRNA and miRNA Target Gene Prediction and Functional Prediction

miRanda and Targetscan were utilized to further explore the target genes of the differentially expressed tsRNAs and miRNAs. Gene Ontology (GO) (http://www.geneontology.org/, accessed on 17 September 2019) analysis aimed to obtain the relationship between the target genes and functional terms. The Kyoto Encyclopedia of Genes and Genomes (KEGG) database (http://www.genome.jp/kegg/, accessed on 17 September 2019) was used for pathway analysis, to find the associated pathways affected by the target genes. For both GO and pathway analysis, *p* < 0.05 was considered as statistically significant.

### 4.7. Statistical Analysis

The read counts in this study were normalized to counts per million mapped reads (CPM). The ‘EdgeR’ R package was used for differential expression analysis. Differentially expressed miRNAs and tsRNAs were identified through fold-change screening (fold change ≥1.5 and *p* < 0.05). SPSS version 22.0 (SPSS Inc., Chicago, IL, USA) was used for the statistics. Data were expressed as mean ± SEM and compared using the unpaired Student’s *t*-test. *p* < 0.05 was considered statistically significant.

## Figures and Tables

**Figure 1 ijms-24-10323-f001:**
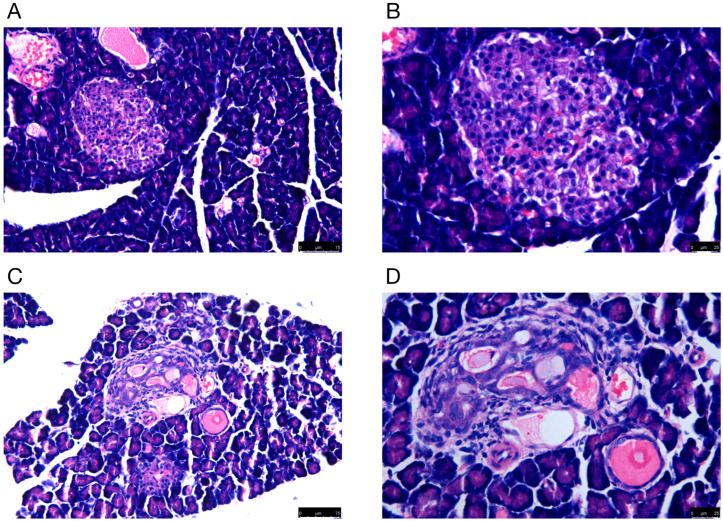
The H&E staining of pancreas in (**A**,**B**) normal and (**C**,**D**) model groups. Representative histology images in each group are exhibited. (**A**,**C**) H&E staining; magnification ×100; scale bars = 75 μm; (**B**,**D**) H&E staining; magnification ×200; scale bars = 25 μm.

**Figure 2 ijms-24-10323-f002:**
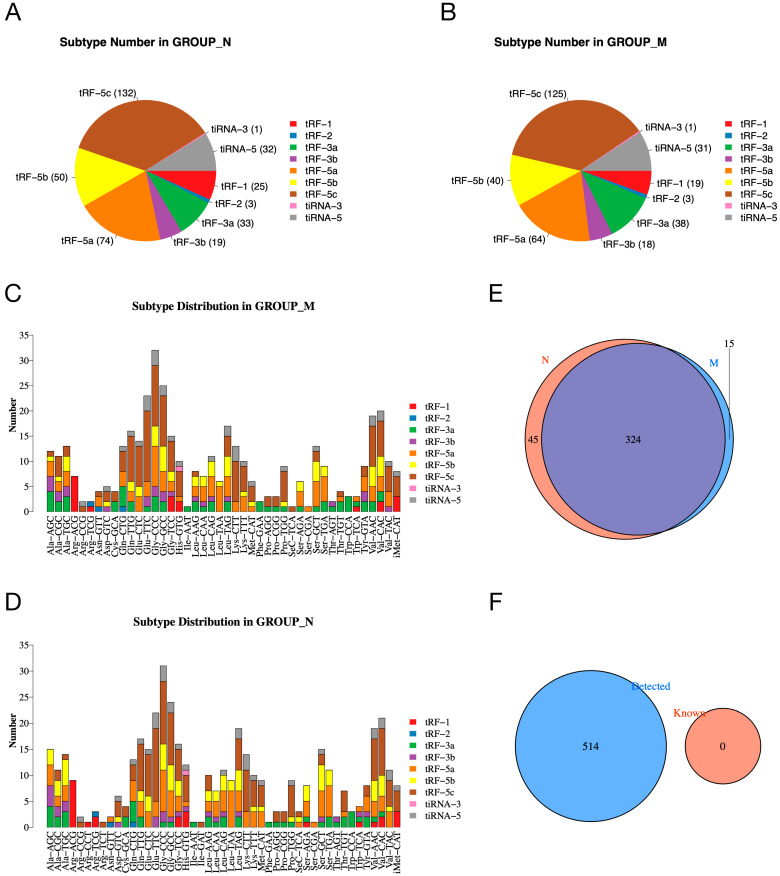
The expression profiles of tsRNAs in normal and model groups. (**A**,**B**) Pie charts of the distribution of subtype tsRNA number in normal (**A**) and model (**B**) groups. (**C**,**D**) Subtype distribution in model (**C**) and normal (**D**) groups. (**E**) Venn diagram shows the number of shared and specifically expressed tsRNAs. (**F**) Venn diagram shows the number of known tsRNAs from tRFdb and the detected tsRNAs in our experiment.

**Figure 3 ijms-24-10323-f003:**
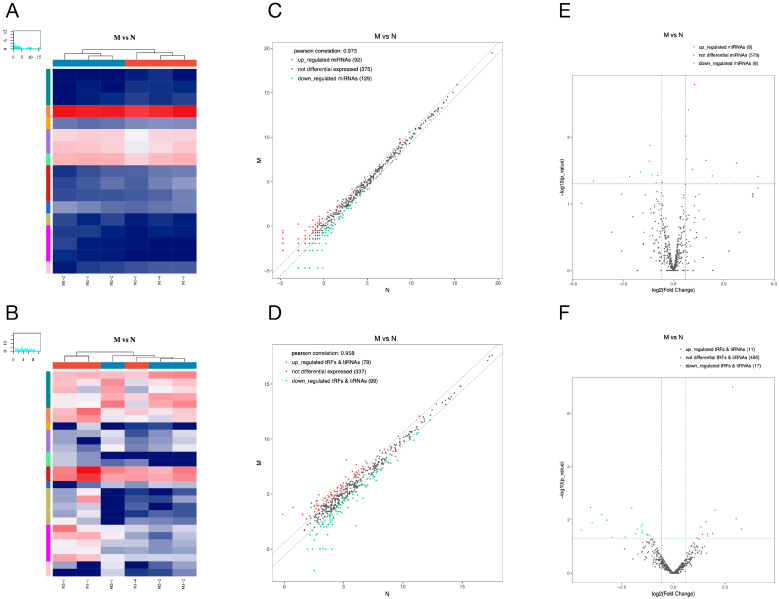
Differentially expressed miRNAs and tsRNAs between model and normal group. The hierarchical clustering heat maps (**A**,**B**), the clatter plots (**C**,**D**) and the volcano plots (**E**,**F**) of differentially expressed miRNAs (**A**,**C**,**E**) and tsRNAs (**B**,**D**,**F**). Detailed description: (**A**,**B**) The color in the panel represents the relative expression level (log2-transformed). The color scale is as follows: blue represents an expression level below the mean, and red represents an expression level above the mean. The colored bar top of the top panel shows the sample group, and the colored bar at the right side of the panel indicates the divisions that were performed using K-means. (**C**,**D**) The CPM values of all miRNAs and tsRNAs are plotted. The values of X and Y axes in the scatter plot are the averaged CPM values of each group (log2 scaled). miRNAs and tsRNAs above the top line (red dots, up-regulation) or below the bottom line (green dots, down-regulation) indicate more than a 1.5-fold change between the two compared groups. Gray dots indicate non-differentially expressed miRNAs and tsRNAs. (**E**,**F**) The values of X and Y axes in the volcano plot are log2 transformed fold change and −log10 transformed *p*-values between the two groups, respectively. Red/green circles indicate statistically significant differentially expressed miRNAs and tsRNAs with a fold change no less than 1.5 and *p*-value ≤ 0.05 (red: up-regulated; green: down-regulated). Gray circles indicate non-differentially expressed miRNAs and tsRNAs, with an FC and/or q-value not meeting the cut-off thresholds.

**Figure 4 ijms-24-10323-f004:**
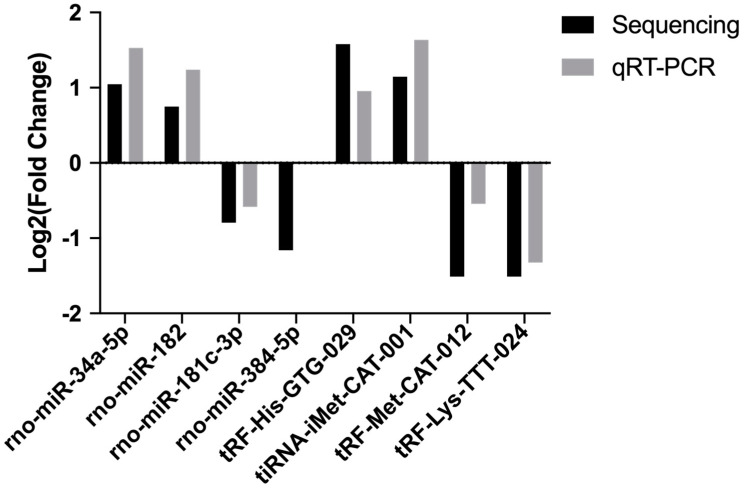
The qRT-PCR validation of selected miRNAs and tsRNAs.

**Figure 5 ijms-24-10323-f005:**
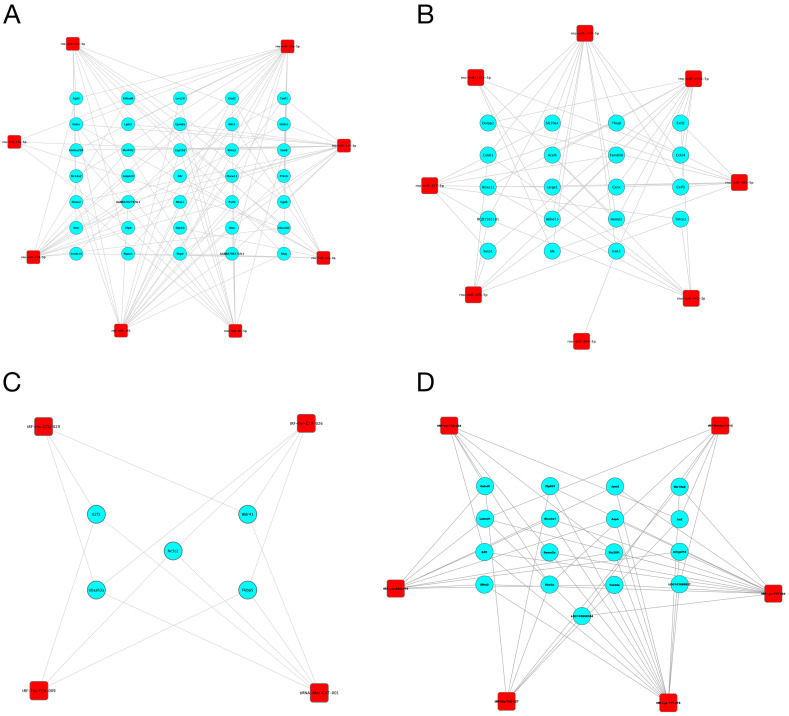
The interaction network of identified miRNAs/tsRNAs and their target genes. (**A**,**B**) The interaction network of up-regulated (**A**) and down-regulated (**B**) miRNAs and their target genes. (**C**,**D**) The interaction network of up-regulated (**C**) and down-regulated (**D**) tsRNAs and their target genes. Red squares represent the miRNAs/tsRNAs; blue circles represent the target genes.

**Figure 6 ijms-24-10323-f006:**
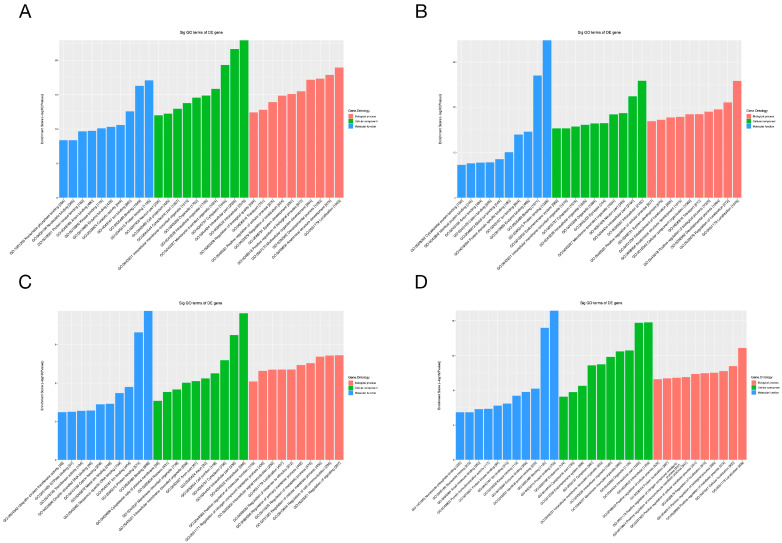
GO analyses of the altered target genes. (**A**–**D**) The GO enrichment score [−log 10 (*p*-value)] analysis of the altered target genes of the down-regulated miRNAs (**A**), up-regulated miRNAs (**B**), down-regulated tsRNAs (**C**) and up-regulated tsRNAs (**D**), showing the top 10.

**Figure 7 ijms-24-10323-f007:**
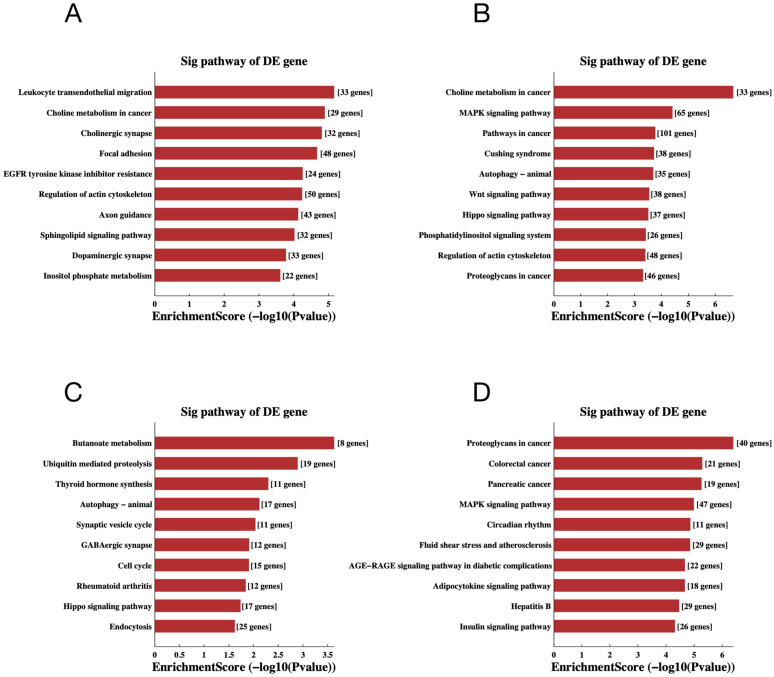
The KEGG pathway analyses of the altered target genes. (**A**–**D**) The KEGG pathway analysis of the altered target genes of the down-regulated miRNAs (**A**), up-regulated miRNAs (**B**), down-regulated tsRNAs (**C**) and up-regulated tsRNAs (**D**). The abscissa is the EnrichmentScore [−log 10 (*p*-value)]. The ordinate is the description of the corresponding pathway of the top 10.

**Table 1 ijms-24-10323-t001:** The FBG levels in each group (*n* = 10) (mean ± S.D.).

Group	FBG Level (mmol/L)	*p*
Normal	4.47 ± 0.65	*p* < 0.0001
Model	15.61 ± 3.29	

**Table 2 ijms-24-10323-t002:** The top five up- and down-regulated miRNAs in model group compared with normal group.

miRNA	Length	Regulation	Fold_Change	*p*_Value
rno-miR-19a-5p	20	up	18.412984	0.0384746
rno-miR-138-2-3p	21	up	8.7834281	0.024325
rno-miR-34a-3p	22	up	3.8157119	0.0380204
rno-miR-542-5p	22	up	3.0840682	0.0222089
rno-miR-34a-5p	22	up	2.069457	0.0016151
rno-miR-3594-5p	21	down	0.063501	0.0453837
rno-miR-448-3p	21	down	0.2210338	0.0390464
rno-miR-369-5p	22	down	0.3206308	0.0330345
rno-miR-137-3p	23	down	0.4342942	0.0228723
rno-miR-384-5p	23	down	0.4461936	0.013194

**Table 3 ijms-24-10323-t003:** The top five up- and down-regulated tsRNAs in model group compared with normal group.

tsRNA	Type	Regulation	Length	Fold_Change	*p*_Value
tRF-Arg-ACG-007	tRF-3a	up	18	9.8496924	0.022108267
tRF-Thr-AGT-013	tRF-3a	up	17	8.217405	0.0091561
tiRNA-His-GTG-002	tiRNA-3	up	40	7.3252156	1.00809 × 10^−07^
tRF-Gly-GCC-018	tRF-3b	up	22	4.042935	0.004316473
tRF-Asn-GTT-031	tRF-2	up	14	3.7170533	0.032505749
tRF-Ser-GCT-008	tRF-5c	down	28	0.0451683	0.024084927
tRF-Trp-TCA-001	tRF-3b	down	21	0.0616459	0.003398912
tRF-Trp-TCA-002	tRF-3b	down	20	0.0650623	0.013008669
tRF-His-GTG-037	tRF-2	down	14	0.0895452	0.006455851
tRF-Arg-TCG-016	tRF-2	down	14	0.1075931	0.010376773

**Table 4 ijms-24-10323-t004:** miRNA and tsRNA primers for qRT-PCR.

Primer Name	Sequence
rno-miR-34a-5p	F:5′GGGGTGGCAGTGTCTTAGC3′ R:5′GTGCGTGTCGTGGAGTCG3′
rno-miR-181c-3p	F:5′GGACCATCGACCGTTGAG3′ R:5′GTGCGTGTCGTGGAGTCG3′
rno-miR-182	F:5′GGCTTTGGCAATGGTAGAAC3′ R:5′GTGCGTGTCGTGGAGTCG3′
rno-miR-384-5p	F:5′GGGGGTTGTAAACAATTCCTAG R:5′GTGCGTGTCGTGGAGTCG3′
tRF-His-GTG-029	F:5′GATCGCCGTGATCGTATAGTG3′ R:5′TCTTCCGATCTACGCAGAGTACTA3′
tiRNA-iMet-CAT-001	F:5′CGATCAGCAGAGTGGCGCAG3′ R:5′ATCTGGGCCCAGCACGCTT3′
tRF-Met-CAT-012	F:5′TCCGACGATCAGTAAGGTCA3′ R:5′GATCTGGCCCGATAGCTTAG3′
tRF-Lys-TTT-024	F:5′ACGATCGCCCGGATAGCT3′ R:5′CCGATCTTGATGCTCTACCGA3′

## Data Availability

The data used to support the findings of this study are available from the corresponding author upon request.

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
