# Peer review of "Small RNA Sequencing Analysis of STZ-Injured Pancreas Reveals Novel MicroRNA and Transfer RNA-Derived RNA with Biomarker Potential for Diabetes Mellitus"

_ijms, 2023, doi:10.3390/ijms241210323_

Round 1

Reviewer 1 Report

Summary: In this manuscript, the authors studied the expression of miRNAs and transfer RNA-derived fragments in type 2 diabetes mellitus(T2DM). The authors have successfully established the mice model for T2DM and performed small RNA sequencing. The results showed that 17 miRNAs and 28 transfer RNA-derived fragments are significantly differentially regulated between T2DM and the control group. Finally, the authors have predicted the target genes for those altered miRNAs and transfer RNA-derived fragments. The results are interesting, especially the novel role of transfer RNA-derived fragments in T2DM, but it needs more experimental evidence to show their role in T2DM.

Comments:

1.    The authors have validated the small RNA sequencing data by qPCR in Figure 4. The authors should discuss and also show evidence that they are amplifying tRNA-derived fragments not the full length tRNA. The northern blot is more appropriate to show that tRNA derived fragments are de-regulated while full length tRNA remain unchanged.

2.    It is important to show the target validation by western blot analysis.

3.    The authors should put more detailed information in some of their Figure legends.

4.    It is unclear how these 17 miRNAs and 28 tRNA-derived fragments are involved in T2DM.

5.    Finally, it is essential to show a rescue experiment to prove their hypothesis that, indeed these miRNAs and tRNA derived fragments are involved in regulating T2DM.

Reviewer 2 Report

I have read and analyzed the manucript of Mo and coauthors. I think that the theme of the manuscript is interesting, but contains crucial mistakes.

1. The first crucial problem is non-standard T2DM model. I know many articles about T2DM modeling, but just clear high fat diet during different times is the SINGLE adequate and physiological model of T2DM. This is a consensuns of all modern diabetologic society all around the world. Any using of STZ is a problem for the clear T2DM modeling.

2. The significant problem both for Introduction and Discussion - in these subsection any comparisons with human data are absent.

3. Why authors used the 45% fat in diet? 60% is the more appropiate approach.

4. Authors did not performed ITT and GTT. Authors can not conclude about the development of type 2 diabetes development. The best possible model - type 1 diabetes, because animals have hyperglycemia and histology of b-cells failure. Authors just can discuss type 1 diabetes with the clarification of conditions in Limitations subsection.

5. Which correction authors used for the multiple comparisons correction? It is critical for genetic research.

I think that this manuscript can be considered for publication just after great revision.

Round 2

Reviewer 1 Report

The authors have addressed all the questions raised. The manuscript is now ready for publication. 

Author Response

We thank for the reviewer's work.

Reviewer 2 Report

Comments 2, 3 and 5 have satisfied me. However, I do not agree with comment 1 and 4. The presented reference are not so fresh and since early 2000s the distinct viewpoint on diabetes modeling has been developed. Nevertheless, the presented data are interesting and can be beneficial both for physicians and basic scientists. As a compromise, my suggestion is the changing of the title on "Small RNA sequencing analysis of STZ-injured pancreas reveals novel microRNA and transfer RNA-derived RNA with biomarker potential for diabetes mellitus" and include respective modification in the manuscript. Otherwise, I can not agree with publication of this data with such description.

Author Response

We thank for the reviewer's suggestion again. And we have revised the manuscript.

Round 3

Reviewer 2 Report

Ok, thanks to authors